# Filtering Bleb Characteristics in Combined Cataract Surgery with Ex-PRESS Implant vs. Non-Penetrating Deep Sclerectomy. A Prospective, Randomized, Multi-Center Study

**Alfonso Anton** [1,†] , **Marcos Muñoz** [1,*,†] , **Marta Castany** [2] , **Alfonso Gil** [3] , **Alberto Martinez** [3] , **Francisco Muñoz-Negrete** [4] , **Jose Urcelay** [5] **and Javier Moreno-Montañes** [6]

1   Department of Glaucoma, Insitut Català de Retina, Universitat Internacional of Catalunya, 08017 Barcelona, Spain
2   Department of Ophthalmology, Hospital Vall d'Hebron, 08035 Barcelona, Spain
3   Department of Ophthalmology, Hospital San Eloy, 48902 Barakaldo, Spain
4   Department of Ophthalmology, Hospital Ramón y Cajal, 28034 Madrid, Spain
5   Department of Ophthalmology, Hospital General Universitario Gregorio Marañon, 28010 Madrid, Spain
6   Department of Ophthalmology, Clínica Universidad de Navarra, 31008 Pamplona, Spain
\*   Correspondence: marcosm@icrcat.com
†   These authors contributed equally to this work.

**Abstract:** (1) Background: After filtering surgery, bleb morphology is an indicator of the factors that may determine the final intraocular pressure (IOP). The present study aimed to evaluate and compare filtering bleb characteristics after combined cataract and glaucoma surgery. (2) Methods: We conducted a prospective multi-center randomized trial. Eyes with glaucoma and cataract were randomly assigned to either phacoemulsification and filtration surgery with an EX-PRESS implant (Alcon) or non-penetrating deep sclerectomy (NPDS) with an ESNOPER implant (AJL). The bleb characteristics were assessed with the Moorfields bleb grading system at months 1 and 12 of follow-up, and the relationship with IOP was analyzed. (3) Results: There were significant changes in bleb appearance between the assessments at month 1 and month 12. The changes in bleb appearance were more evident in the EX-PRESS group. The central area in the EX-PRESS group decreased from 2.9 at month 1 to 2.4 at month 12 ($p = 0.014$). Bleb height in the EX-PRESS group decreased from 2.3 at month 1 to 1.8 at month 12 ($p = 0.034$). The vascularity of the central area in the NPDS group decreased from 1.8 at month 1 to 1.3 at month 12 ($p = 0.02$). The maximal bleb area was inversely related ($r = -0.39$; $p = 0.03$) to the IOP in the NPDS group at month 1. Vascularity in the central area was directly related ($r = 0.39$; $p = 0.01$) to a higher IOP in the EX-PRESS group at month 1. Vascularity in the central area ($r = 0.56$; $p < 0.001$) and maximal area ($r = 0.37$; $p = 0.012$) at month 1 was directly related to a higher IOP in the EX-PRESS group at month 12. (4) Conclusions: More intense vascularity at month 1 was related to a higher final IOP in the EX-PRESS group. Larger blebs were associated with a lower IOP in the NPDS group.

**Keywords:** bleb grading; filtering bleb; EX-PRESS shunt; deep sclerectomy

## 1. Introduction

The EX-PRESS glaucoma filtration device and non-penetrating deep sclerectomy (NPDS) are alternatives to traditional trabeculectomy that aim to reduce postoperative complications while maintaining the intraocular pressure (IOP) lowering effect [1–3]. Both procedures share certain aspects of the surgical technique: a scleral flap, an implant placed under the flap, and both produce a visible filtering bleb.

After the initial postoperative period, aqueous humor resistance, and therefore IOP, is determined mainly by the size and the walls of the subtenon/episcleral space where the filtering bleb is forming. Bleb morphology and characteristics are indicators that

may significantly influence the final IOP after surgery and help to identify postoperative progress and complications. Since bleb morphology evolves over time, it is necessary to assess and register these changes, and to adapt postoperative management to them. Different systems have specifically been designed to evaluate the filtering bleb in clinical practice [4–7]. Some of these systems only describe the appearance or vascularization globally, without differentiating between the central and peripheral zone. Thus, these systems fail to describe blebs with mixed characteristics (e.g., a cystic center with a diffuse peripheral area). For these reasons, the Moorfields bleb grading system (MBGS) [6], which evaluates area, height, and vascularity, as well as differences between the central and peripheral bleb zone, was chosen for this study.

The purpose of this study was to assess and compare bleb morphology in two filtering procedures (Phaco-Ex-PRESS vs. Phaco-NPDS) in order to identify morphologic features that could help in the early detection of cases with a higher risk of failure. The association of bleb morphology or bleb characteristics with IOP values was also assessed.

## 2. Materials and Methods

This was a prospective, randomized, multi-center study. Institutional Review Board/ Ethics Committee approval was obtained at each participating center. This research followed the tenets of the Declaration of Helsinki. Written informed consent was obtained from all participants before their inclusion in the study. The trial was registered at http://clinicaltrials.gov (NCT03201354, accessed on 19 January 2023).

The sample size was calculated accepting an alpha risk of 0.05 and a beta risk of 0.2 (statistical power of 80%). Forty-six subjects were estimated to be necessary for each group in order to recognize a statistically significant difference $\geq 3$ mmHg in the average postoperative IOP between the groups. A detailed description of the sample calculation, randomization, the examinations, the surgical procedure, and the postoperative care, has been reported elsewhere [8]. The sample of 98 patients recruited for the study allowed us to identify differences of 0.5 in the bleb height between the groups with a statistical power of 90% and to identify differences of 1 in the central vascularization of the bleb between the groups with the same statistical power of 90%.

Persons diagnosed with primary open-angle glaucoma, pigmentary or pseudo-exfoliation glaucoma, and cataracts were included in the study. The inclusion criteria was as follows: aged older than 18 years; disease that is poorly controlled with medical treatment at the discretion of the ophthalmologist (i.e., progression, IOP higher than target IOP, poor adherence); a glaucomatous visual field (VF) [8]; a glaucomatous optic nerve; and optical coherence tomography (OCT) outside the normal limits and open-angle. The exclusion criteria consisted of the following: persons with inflammatory, neovascular, closed-angle, normotensive, or steroid-induced glaucoma; a mean deviation of <20 dB in VF; any previous incisional glaucoma surgery; other eye surgery within the year previous to enrolment; myopia $\leq -6$ dioptres; hyperopia $\geq +5$ dioptres; or unwillingness to participate.

All the participants underwent phacoemulsification and EX-PRESS filtering device P-50 model (Alcon Laboratories, Fort Worth, TX, USA) (50 eyes), or phacoemulsification and NPDS with an ESNOPER (AJL Ophthalmics, Vitoria, Spain) implant (48 eyes) after being randomized to one of two groups. Seven experienced surgeons in six centers performed the same standardized surgical technique. The surgical procedure consisted of phacoemulsification through a clear corneal incision (2.1–2.8 mm). In all cases, the filtering procedure consisted of a fornix-based conjunctival flap. Mitomycin (MMC) 0.2 mg/mL was used intraoperatively in both groups. The scleral flap was squared in both groups ($5 \times 5$ mm $^1/_3$ of depth in the NPDS and $4 \times 4$ mm $\frac{1}{2}$ of depth in the EX-PRESS group). The EX-PRESS device was placed according to the manufacturer's instructions, and the position of the implant in the anterior chamber was verified. In the NPDS group, a deep scleral flap ($4 \times 4$ mm) was dissected, and an Esnoper V2000 model (AJL) implant was positioned under the superficial scleral flap.

Postoperative care: Participants were evaluated preoperatively and at day 1, weeks 1, 2, and 3, and months 1, 3, 6, and 12 after the intervention. At each visit, the examination included the measurement of BCVA, the measurement of IOP with applanation tonometry [9], a slit-lamp examination, and the registration of adverse events and complications. The subjects in both groups underwent the same postoperative treatment: moxifloxacin drops (every 6 h for 1 week) and dexamethasone (every 2 h for 1 month, every 4–6 h during the second month, tapering off during the third month). The investigators assessed and performed suture lysis, goniopuncture, and subconjunctival injections of anti-fibrotic agents and bleb needling when it was considered necessary. These treatments were considered interventions to maintain the flow of the surgical procedure, and not a failure of the surgery.

Six observers (all ophthalmologists and glaucoma specialists) evaluated the blebs during the slit-lamp examinations at the 1- and 12-month postoperative visits using the Moorfields bleb grading system reference photographs [6]. The Moorfields bleb grading system evaluates the central and maximal area, the height, the central, peripheral, and non-bleb vascularity, and the presence of subconjunctival blood. The parameters of the Moorfields system are summarized in Table 1.

**Table 1.** Moorfields bleb grading system parameters.

| Parameter | Symbol | Range | Represents | Notes |
|---|---|---|---|---|
| Central area | 1a | 1–5 | 0% of photo area to 100% | Demarcated central area |
| Maximal area | 1b | 1–5 | | Total elevated area |
| Height | 2 | 1–4 | Point of maximum height | |
| Central vascularity | 3a | 1–5 | Avascular (1) to severe (5) | 2 = "normal" |
| Peripheral vascularity | 3b | 1–5 | | |
| Non-bleb vascularity | 3c | 1–5 | | |
| Subconjunctival blood | Scb | 0–1 | Yes or no | |

Statistical analyses were performed using SPSS for Windows, version 21.0 (IBM Corp., Armonk, NY, USA). The values obtained during the bleb assessment were compared between the groups and between months 1 and 12 of follow-up using the Mann–Whitney and Wilcoxon tests. The relationship between the bleb parameters and the IOP was analyzed using the Pearson–Spearman test. The main outcome measures were the mean IOP and the bleb morphologic features based on the Moorfields bleb grading system.

### 3. Results

Fifty eyes were assigned to the Phaco-Ex-PRESS group and forty-eight eyes were assigned to the Phaco-NPDS group. Ninety-five participants (97%) aged 54 to 89 years completed the 12-month follow-up. The two groups were similar in terms of age, gender, ethnicity, type of glaucoma, visual acuity, preoperative IOP, and the number of glaucoma medications used at the moment of surgery [10].

*3.1. Intraocular Pressure*

At one year of follow-up, the IOP was significantly reduced from baseline in both groups ($p < 0.05$) (26.4% in the EX-PRESS group and 30% in the NPDS group). At the end of the study, the mean IOPs ($\pm$SD) were 13.9 ($\pm$3.3) and 13.3 ($\pm$3.6) mmHg in the EX-PRESS and NPDS groups, respectively, with no significant differences between the two groups (Figure 1).

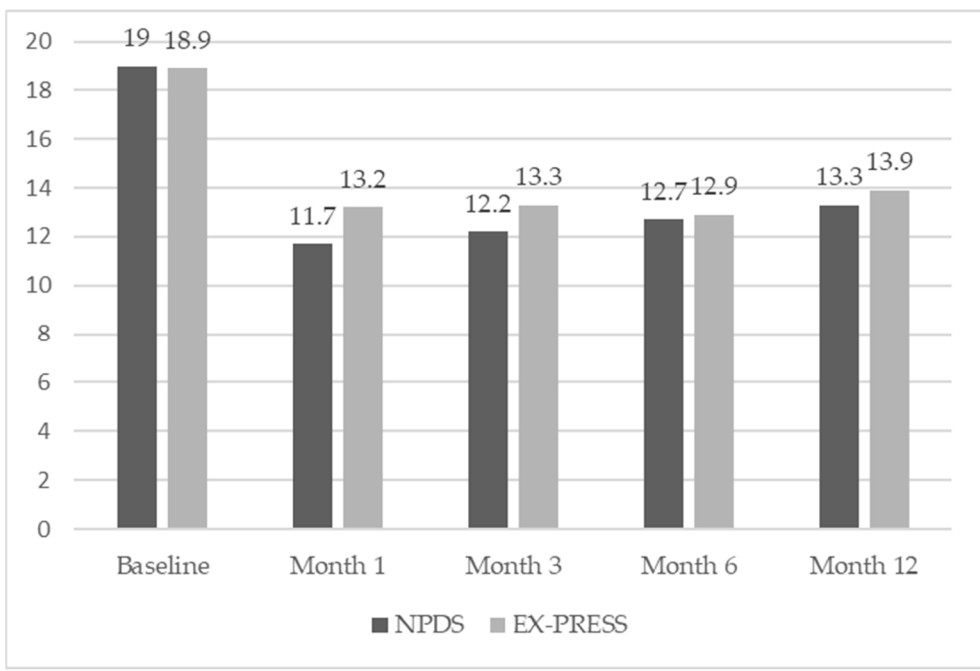

**Figure 1.** Mean IOP at baseline and follow-up visits in the NPDS vs. EX-PRESS groups in mmHg. The study was completed by 47 persons in the NPDS group and 48 persons in the EX-PRESS group. The mean IOP was similar in the two groups at all time points.

*3.2. Postoperative Interventions*

Both of the groups received postoperative interventions if needed during the follow-up period (Table 2). There were 59 and 26 interventions in the EX-PRESS and NPDS groups, respectively. The total number of patients who received interventions was 37 versus 24 in the EX-PRESS versus NPDS groups, respectively ($p = 0.01$). All the interventions were performed at the slit lamp in all cases. No patient required a glaucoma re-operation during the follow-up period.

**Table 2.** Postoperative interventions in the EX-PRESS (n = 50) and NPDS (n = 48) groups after combined surgery during the 1-year follow-up period.

| Intervention | EX-PRESS | | NPDS | | | |
|---|---|---|---|---|---|---|
| | N Procedure * | N Subjects * | N Procedure * | N Subjects * | *p* ** | RR 95% CI |
| 5 Fluorouracil | 15 | 8 | 3 | 3 | 0.20 | 2.56 (0.72–9.08) |
| Goniopuncture | 0 | 0 | 17 | 15 | **<0.001** | |
| Mitomycin C | 1 | 1 | 0 | 0 | 1 | |
| Needling | 12 | 9 | 4 | 4 | 0.16 | 2.16 (0.71–6.54) |
| Suture Lysis | 31 | 19 | 2 | 2 | **<0.001** | 9.12 (2.24–37.06) |
| Total number of interventions | 59 | **37** | 26 | **24** | **0.01** | |

NPDS = non-penetrant deep sclerectomy. * first column: the number of interventions, second column: the number of subjects who underwent the intervention. ** Chi-square test.

*3.3. Bleb Assessment*

The bleb parameters that were assessed with the Moorfields grading system are shown in Tables 3 and 4. There were no statistically significant differences between the groups at months 1 and 12 of follow-up in any of the parameters assessed. There were significant changes in the appearance of the bleb when comparing the assessments at month 1 and month 12 (Table 3).

**Table 3.** Moorfields bleb grading system parameters in both groups.

| EX-PRESS | Month 1 [1] | Month 12 [1] | $p$ [2] |
|---|---|---|---|
| Central area (1–5) | 2.9 ± 1.0 | 2.4 ± 0.9 | **0.01** |
| Maximal area (1–5) | 3.16 ± 0.9 | 3.09 ± 1.1 | 0.73 |
| Height (1–4) | 2.3 ± 0.7 | 1.8 ± 0.9 | **0.03** |
| Central vascularity (1–5) | 2.1 ± 1.0 | 1.5 ± 0.5 | **0.005** |
| Peripheral vascularity (1–5) | 2.3 ± 0.9 | 1.8 ± 0.4 | **0.005** |
| Non-bleb vascularity (1–5) | 1.86 ± 1.0 | 1.6 ± 0.6 | 0.12 |
| **NPDS** | **Month 1** | **Month 12** | $p$ (a) |
| Central area (1–5) | 2.8 ± 0.9 | 2.7 ± 1.2 | 0.61 |
| Maximal area (1–5) | 3.5 ± 0.8 | 3.55 ± 1.1 | 0.85 |
| Height (1–4) | 2.1 ± 0.5 | 2.0 ± 0.7 | 0.25 |
| Central vascularity (1–5) | 1.8 ± 0.7 | 1.3 ± 0.5 | **0.02** |
| Peripheral vascularity (1–5) | 2.1 ± 0.8 | 1.8 ± 0.7 | 0.10 |
| Non-bleb vascularity (1–5) | 1.52 ± 0.7 | 1.5 ± 0.6 | 0.76 |

NPDS: non-penetrating deep sclerectomy. [1] Average and standard deviation. [2] Wilcoxon test.

**Table 4.** Comparison of the Moorfields bleb grading system parameters between the NPDS and EX-PRESS groups.

| Parameter | Month 1 [1] ($p$ Value) | Month 12 [1] |
|---|---|---|
| Central area | 0.567 | 0.311 |
| Maximal area | 0.164 | 0.153 |
| Height | 0.523 | 0.34 |
| Central vascularity | 0.15 | 0.19 |
| Peripheral vascularity | 0.271 | 0.553 |
| Non-bleb vascularity | 0.188 | 0.435 |

[1] Mann–Whitney test.

Table 3 Legend: most of the bleb parameters changed significantly in the EX-PRESS group, whereas only central vascularity decreased significantly in the NPDS group at months 1 and 12 of follow-up.

Table 4 Legend: No significant differences were found when comparing the bleb parameters at months 1 and 12 for all study participants. For the group-specific differences, see Table 3.

The changes in bleb appearance were more evident in the EX-PRESS group than in the NPDS group. In the NPDS group, only central-area vascularity showed a significant decrease from 1.8 to 1.3 at months 1 and 12 of follow-up ($p = 0.02$). In the EX-PRESS group, the central area significantly decreased ($p = 0.014$) from 2.9 to 2.4, bleb height significantly decreased from 2.3 to 1.8 ($p = 0.03$), and bleb vascularity also decreased from 2.3 to 1.8 in the central zone and from 2.1 to 1.5 in the peripheral zone ($p < 0.05$) from month 1 to month 12.

Finally, some bleb parameters showed a significant relationship with IOP when comparing the 1- and 12-month postoperative periods. The maximal bleb area of the NPDS group was inversely related to the IOP at 1 month of follow-up (r = −0.39; $p = 0.03$). More intense vascularity in the central area of the EX-PRESS group was directly related to a higher IOP at month 1 of follow-up (r = 0.39; $p = 0.01$). Additionally, in the EX-PRESS group, the vascularity in the central and peripheral areas measured 1 month after the procedure was directly related to a higher IOP at month 12 of follow-up (r = 0.56 ($p < 0.001$) and r = 0.378 ($p = 0.012$), respectively. No subconjunctival blood was observed in either group 1 month or 12 months after the procedure. Figure 2 shows examples of bleb morphology associated with different prognoses.

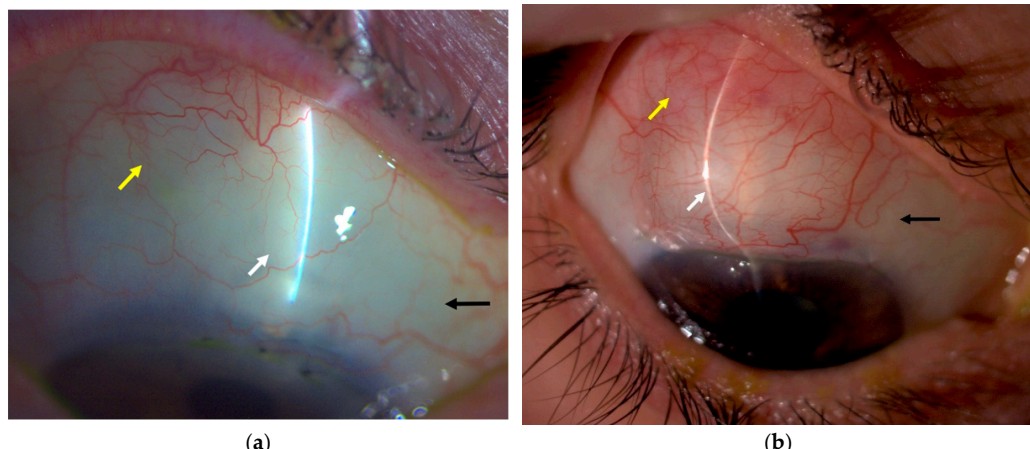

(**a**)                                                             (**b**)

**Figure 2.** (**a**) Good prognosis bleb. Moorfields bleb grading system (MBGS): 1a,b (3), 2 (2), 3a,b,c (2). (**b**) Poor prognosis bleb. MBGS: 1a,b (3), 2 (3), 3a (4), 3b (4), 3c (2). White arrow, central zone; yellow arrow, maximal area zone; black arrow, non-bleb zone.

## 4. Discussion

The present study evaluated bleb appearance using a standardized grading system and assessed IOP in two different filtering procedures performed in combination with phacoemulsification (EX-PRESS vs. NPDS). To our knowledge, no previous studies have compared the morphology of filtering blebs formed after EX-PRESS and NPDS.

At 12 months of follow-up, the IOP was significantly reduced in both groups (26.4% in the EX-PRESS group and 30% in the NPDS group), which seems less intense than the results obtained in other studies [11–14]. This is probably because in our study, no washout period was performed before the baseline IOP measurement, as suspending the hypotensive treatment could have increased the participants' risk of progression.

Although our results do not demonstrate significant differences in the bleb parameters between the two groups at any study point, there were some group differences in the changes in bleb appearance. The characteristics of the blebs evolved slightly differently in the two groups as the scarring process advanced. Changes observed throughout the follow-up period were more marked and statistically significant in the EX-PRESS group than in the NPDS group. A comparison of EX-PRESS blebs between months 1 and 12 revealed that the central area decreased, the bleb height lowered, and the central and maximum vascularization also decreased. In contrast, the NPDS group showed only a significant decrease in central vascularization, and the NPDS blebs showed a non-significant tendency to be lower, more diffuse, and less vascularized than the EX-PRESS blebs. A possible explanation for these differences in the characteristics and the postoperative course could be that the aqueous humor flow in NPDS is less intense, more diffuse, and homogeneous due to the limitation of flow in the trabeculo-descemetic membrane. The finding that EX-PRESS blebs tended to be more elevated and prominent in the first weeks or months after surgery could be due to the more accentuated and localized aqueous flow, only limited by the interior diameter of the implant and the scleral flap sutures, when compared to the slow diffuse flow occurring through the NPDS membrane.

At 1 year of follow-up, the blebs were less elevated in the EX-PRESS group than in the NPDS group; however, this difference was not statistically significant (1.88 versus 2.03 NPDS, $p = 0.33$). There was a non-significant change of $-7\%$ in the heights of the NPDS blebs between months 1 and 12 ($p = 0.25$), whereas the mean height decreased in the EX-PRESS group by $-18.3\%$ ($p = 0.03$). Good and Kahook [15] compared the characteristics of filtering blebs between EX-PRESS and trabeculectomy and found less vascularization and more diffused blebs in the EX-PRESS group in the early postoperative period, but these differences disappeared at the end of 2 years of follow-up. Although their results may seem contradictory to ours, it is possible that a slower and more homogeneous the flow results in less hyperemia in the filtering blebs, leading to a more diffuse appearance. If

this theory is correct, their results and ours are congruent, and the lowest and more diffuse flow through the NPDS membrane may result in the formation of less vascular and more diffuse filtering blebs. It can be speculated that the type of flow through the NPDS better resembles physiological flow through the trabecular meshwork.

IOP correlated with some bleb parameters. The maximum area of blebs in the NPDS group was inversely correlated with IOP at month 1. This means that a larger area implies a lower IOP, and vice versa. Additionally, the vascularization in the EX-PRESS group was significantly associated with a higher IOP at month 1. Perhaps the most interesting finding was observed in the direct correlation (r = 0.56; *p* > 0.001) between central and maximum vascularization at month 1 and a higher IOP at month 12 in the EX-PRESS group. Specifically, the greater the vascularization in the early postoperative period, the higher the IOP at the end of the study. This finding confirms previous knowledge indicating that a less vascularized bleb is desirable in order to obtain better IOP lowering, and supports the need for dedicated and individualized care of the filtering bleb. Additionally, our results suggest that the filtration bleb with an EX-PRESS implant is possibly evidence of a more active scarring/modelling process, and also could explain why it requires more intense postoperative management [10] compared with the more stable-looking NPDS blebs.

The appearance of NPDS blebs was probably also influenced by the multiple outflow path proposed for this surgical technique. It has been proposed that once the aqueous humor passes through the trabeculo-descemetic membrane, it can filter out to the open endings of the Schlemm channel, the intrascleral space, the suprachoroidal space, or the subconjunctival space. It is accepted that the flow in the NPDS is not as dependent on the filtering bleb [16], unlike trabeculectomy and EX-PRESS. It has been reported that up to 50% of patients who have undergone NPDS surgery may not have a visible filtering bleb, but still have an effective intervention [17]. However, in our study, all the patients who underwent NPDS surgery developed a visible conjunctival bleb. Ultrabiomicroscopy (UBM) has shown that a slightly elevated and diffuse bleb can be found even several years after the NPDS intervention and that these blebs tend to be flatter than those observed in trabeculectomy [18].

Suture lysis is frequently used in the postoperative period of filtration surgery, particularly in penetrating surgery, to increase the flow of aqueous humor through the borders of the scleral flap. Suture lysis is usually indicated when intraocular pressure is not controlled after massage of the bleb, when the IOP is above the target, or when a flat bleb is observed. Suture lysis can be useful until months after surgery if MMC is used intraoperatively [19–21]. In our study, as expected, most of the suture lyses were performed in the Ex-PRESS group, where the flow through the scleral flap is more sensitive to this procedure than in NPDS.

Conjunctival and/or scleral scarring is one of the main limiting factors in the success of filtering surgery. This process involves an inflammatory reaction with fibroblast proliferation and extracellular matrix production. Since its introduction, 5-Fluorouracil and Mitomycin have been used to modulate the scarring process [22].

Currently, 5-FU is used in the postoperative period with three to five injections of 0.1 mL 50 mg/mL every 24–72 h. The postoperative use of 5-FU is an effective strategy to improve the outcomes of filtering surgery. In the present study, we used 5-FU in eight and three patients in the EX-PRESS and NPDS groups, respectively. The use of Mitomycin-C (MMC) was described in 1983. The most common method of using MMC is intraoperatively. Nevertheless, MMC can be used in the postoperative period to treat blebs with a higher risk of scarring. The postoperative dose of MMC is 0.01 mL 0.02 mg/mL MMC. We treated one case with a postoperative subconjunctival MMC in the EX-PRESS group.

When analyzing the bleb characteristics of the eyes treated with 5-FU vs. those that were not, we found a non-significant tendency for lower, more vascularized, and smaller blebs among those treated with the anti-fibrotic agent. This fact was expected since the indication of the 5-FU injections decision was left to the surgeons, who agreed before the study that this treatment would be applied in blebs with a higher risk of failure. The lack of significance in this analysis was inevitable because the number of eyes requiring 5-FU was minimal.

Although the use of anti-fibrotic agents [23] is the main factor associated with the production of avascular blebs, greater flow of aqueous humor is also related to the formation of avascular blebs [24]. In our study, there was only one case of an avascular bleb in the EX-PRESS group and none in the NPDS group. Nevertheless, a longer follow-up would be advisable in order to confirm the frequency of this event in both groups.

In our study, the method used to assess the blebs was a direct comparison during the slit-lamp examination, using reference photographs from the Moorfields bleb grading system. Previous studies have shown good levels of interobserver agreement, variability, and reproducibility [6,25,26] for this grading system. In the present study, the blebs were assessed by one observer in each center (glaucoma specialist). Some authors advocate using photographs of the bleb to evaluate its morphology. We did not find this approach feasible because some of the participating centers had difficulties in obtaining good quality standardized bleb photographs and because it is very difficult to assess bleb height in 2D photographs.

Among the strengths of this study are its prospective, randomized, and multi-center design, together with the use of a freely available and previously tested method to assess blebs [6,15,25]. Nevertheless, this study had certain limitations, which we believe did not prevent us from reaching our objectives. First, even though 1 year is a reasonable follow-up time [9], a longer follow-up period would be interesting to detect further changes in bleb morphology and IOP. Second, more frequent bleb assessments could have shown more detailed descriptions of changes in the blebs during the early postoperative period. Additionally, interobserver agreement could not be assessed since only one evaluator graded the blebs in each center. Third, anterior segment OCT (AS-OCT) was not used for bleb assessment. Since the introduction of OCT technology, various studies [27–32] have described bleb morphology based on AS-OCT. It certainly adds information about the internal structure of the bleb (e.g., the degree of light reflected and patterns), but also has major limitations when two different surgical techniques are compared over time. First, the measure of any bleb parameter is highly dependent on the particular zone scanned. It is challenging to acquire the scan in the same place each time the bleb is assessed at different time points, limiting the reproducibility of measurements and their comparison over time. Second, the same bleb frequently shows different morphology patterns when distinct zones are examined. Finally, there is no consensus regarding the characteristics that serves as biomarkers for surgical success, nor regarding recommendations of the role of AS-OCT for standardized bleb assessment [33] or its role in postoperative bleb management.

## 5. Conclusions

Our study shows slight but significant differences in the appearance of, and changes in, blebs during the follow-up period between the groups. More intense vascularity in EX-PRESS blebs in the early postoperative period (month 1) was related to a higher IOP at the end of the study. Larger blebs were associated with a lower IOP (i.e., better IOP lowering) in the NPDS group. The results of this study support the usefulness of detailed assessments of bleb morphology in order to detect changes and to individually adapt postoperative treatment accordingly (e.g., topical steroids and 5-fluorouracil injections) in order to achieve the best bleb performance and lower IOP values.

**Author Contributions:** Conceptualization, M.M., A.A., M.C., A.G., A.M., F.M.-N., J.U. and J.M.-M.; methodology, M.M. and A.A.; validation, M.M., A.A., M.C., A.G., A.M., F.M.-N., J.U. and J.M.-M.; formal analysis, M.M. and A.A.; investigation, A.A., M.C., A.G., A.M., F.M.-N., J.U. and J.M.-M.; data curation, A.A.; writing—original draft preparation, M.M.; writing—review and editing, M.M., A.A. and J.M.-M.; visualization, M.M.; supervision, A.A.; project administration, M.M.; funding acquisition, A.A. All authors have read and agreed to the published version of the manuscript.

**Funding:** Alcon Laboratories (Fort Worth, TX, USA), provided the EX-PRESS devices, and part of the funding. AJL Ophthalmic (Vitoria, Spain) offered a discount for the ESNOPER implants. The Spanish National Health System provided facilities for the surgical procedures.

**Institutional Review Board Statement:** Institutional Review Board/Ethics Committee approval was obtained at each participating center. This research followed the tenets of the Declaration of Helsinki. The trial was registered at http://clinicaltrials.gov (Identifier NCT03201354, accessed on 19 January 2023). We adhered to the 2010 CONSORT statement for the reporting of randomized clinical trials.

**Informed Consent Statement:** Informed consent was obtained from all the subjects involved in the study.

**Data Availability Statement:** The principal investigator, Alfonso Anton-Lopez, had full access to all the data in the study and takes responsibility for the integrity of the data and the accuracy of the data analysis.

**Acknowledgments:** The authors thank Juan Martin from the International University of Catalonia, Barcelona, Spain for his contribution to the statistical analysis.

**Conflicts of Interest:** The authors declare no conflict of interest. The funders had no role in the design of the study; in the collection, analyses, or interpretation of data; in the writing of the manuscript; or in the decision to publish the results. This study was performed as part of the research activities of the Thematic Collaborative Research Network (RedTemática de Investigación Colaborativa) RETICS-OftaRed of the Health Institute of Carlos III, Spain.

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
