# Peer review of "Filtering Bleb Characteristics in Combined Cataract Surgery with Ex-PRESS Implant vs. Non-Penetrating Deep Sclerectomy. A Prospective, Randomized, Multi-Center Study"

_2813-1053, doi:10.3390/jcto1010004_

Round 1

Reviewer 1 Report

The authors evaluated the difference of bleb appearance between Ex-PRESS implantation and NPTS procedure in a multicenter, randomized study. The manuscript is well-written and well-organized. I just have some minor concerns:

1.       In ‘Abstract’, the differences between the two groups should be highlighted in the section ‘Results’ and ‘Conclusion’.

2.       Line 88, do the authors mean ‘myopia ≦-6 diopters’?

Reviewer 2 Report

Dear authors,

you presented an interesting research regarding bleb morphology after two different glaucoma procedures. This is a well-written article which could be considered for publication. In my opinion, both the results and discussion section should be briefly implemented: in particular, have you found any correlation between the rate of post-operative slit-lamp interventions in the ExPRESS group and the morphology or vascularization of the bleb? Since a significative rate of re-interventions have been reported, this element could have affected the vascularization and the area of the bleb.

Moreover, the use of a subjective grading system remains a strong limitation of this study, since OCT analysis could have highlighted much more characteristic and morphological evolution of the blebs.

Finally, some small correction should be perfomed:

1.       Line 97: please adjust the fractions.

2.       Lines 164-174: please consider switching the position of Table 3 and 4, since Table 4 seems to be the report of general findings in the two groups.

3.       Line 247: is it month 1 or month 12? Please correct.

4.       Lines 278-299: since no statistical analysis was performed regarding the use of 5FU instead of other anti-fibrogenic agents, this sections coul be summarized.

5.       Lines 322-324: AS-OCT has been used in a recent paper to compare the morphology of some MIGS-plus procedures. Please, consider adding the reference. https://pubmed.ncbi.nlm.nih.gov/35626405/

6.       Please, perform a review of the language since some typos are still present.

Kind regards
